# Effect of Pre-Induced Mesenchymal Stem Cell-Coated Cellulose/Collagen Nanofibrous Nerve Conduit on Regeneration of Transected Facial Nerve

**DOI:** 10.3390/ijms23147638

**Published:** 2022-07-11

**Authors:** GwangWon Cho, Changjong Moon, Nagarajan Maharajan, Mary Jasmin Ang, Minseong Kim, Chul Ho Jang

**Affiliations:** 1Department of Biology, College of Natural Science, Chosun University, Gwangju 61452, Korea; gwcho@chosun.ac.kr (G.C.); geneticnaga1990@gmail.com (N.M.); 2Department of Life Science, BK21-Plus Research Team for Bioactive Control Technology, Chosun University, Gwangju 61452, Korea; 3Department of Veterinary Anatomy, College of Veterinary Medicine and BK21 FOUR Program, Chonnam National University, Gwangju 61186, Korea; moonc@chonnam.ac.kr; 4College of Veterinary Medicine, University of the Philippines Los Baños, Los Baños 4031, Philippines; ang.maryjasmin@gmail.com; 5Advanced Biomaterial Team, Medical Device Development Center, Daegu-Gyeongbuk Medical Innovation Foundation, Dong-gu 41061, Korea; mkim0301@kmedihub.re.kr; 6Department of Otolaryngology, Chonnam National University Medical School, Gwangju 61469, Korea

**Keywords:** cellulose, collagen, nanofibrous nerve conduit, pre-induced mesenchymal stem cell, nerve regeneration

## Abstract

(1) Objective: In order to evaluate the effect of a pre-induced mesenchymal stem cell (MSC)-coated cellulose/collagen nanofibrous nerve conduit on facial nerve regeneration in a rat model both in vitro and in vivo. (2) Methods: After fabrication of the cellulose/collagen nanofibrous conduit, its lumen was coated with either MSCs or pre-induced MSCs. The nerve conduit was then applied to the defective main trunk of the facial nerve. Rats were randomly divided into three treatment groups (n = 10 in each): cellulose/collagen nanofiber (control group), cellulose/collagen nanofiber/MSCs (group I), and cellulose/collagen nanofiber/pre-induced MSCs (group II). (3) Results Fibrillation of the vibrissae of each group was observed, and action potential threshold was compared 8 weeks post-surgery. Histopathological changes were also observed. Groups I and II showed better recovery of vibrissa fibrillation than the control group. (4) Conclusions: Group II, treated with the pre-induced MSC-coated cellulose/collagen nanofibrous nerve conduit, showed the highest degree of recovery based on functional and histological evaluations.

## 1. Introduction

Facial nerve damage is a frequent complication of traumatic injury that occurs after facial bone trauma or head and neck injury or as a surgical side effect. Among peripheral nerve injuries, facial nerve injury considerably decreases quality of life because facial expression is the most important emotional signal used to convey diverse thoughts, ideas, and emotions. 

To reconstruct the nerve gap, autografting is ideal, but it is associated with problems at the donor site, such as neuroma formation, skin incision-induced scar formation, and lack of sensation. A nerve conduit minimizes these complications. Such conduits are divided into autologous and synthetic nerve conduits. Autologous nerve conduits can be obtained using arteries, veins, and muscle, but donor-site problems are inevitable. Synthetic nerve conduits can be classified as nondegradable and biodegradable [1]. Recently, biodegradable nerve conduits have been developed using collagen, polylactide, polycaprolactone, polyglycolic acid, chitosan, and cellulose. The main advantage of these biodegradable nerve conduits is that they are readily available. Cellulose acetate is an excellent structural material and a rich structural biomolecule used in nature and industry. Because of its good mechanical properties, it is occasionally mixed with gelatin scaffold [2].

Nanofibers are synthetic fibers that naturally mimic the extracellular matrix and provides an environment suitable for cell attachment, differentiation, and proliferation [3]. The application of electrospinning was to mimic the structure of a nerve conduit. 

Adult mesenchymal stem cells (MSCs) can be separated from umbilical cord blood, bone marrow, adipose tissue, and the skin [4]. MSCs can be differentiated into neurons and this property can be utilized as a therapeutic target to promote neuroprotection and neurogenesis. Owing to the absence of precursor cells around damaged areas, regeneration is impossible in these areas. To overcome this limitation, various potent stem cell sources, such as MSCs and neural stem cells, have been used in combination with nerve conduits to treat peripheral nerve defects [5,6,7,8]. However, after transplantation of MSCs into the neural damaged area, only a minority of MSCs can be differentiated into neuronal cells in vivo [9].

To resolve this problem, chemical compounds or growth factors have been used to induce neural differentiation of MSCs [10]. However, according to follow-up studies, morphological and molecular transformation of MSCs may occur as a stress response of the cells rather than their actual differentiation into neuronal cells [10]. During neural differentiation, the endoplasmic reticulum of MSCs is highly susceptible to stress owing to the nature of the differentiation process itself [11]. If the cell overcomes the stress, it shows further adaptation by performing other functions; otherwise, it dies. [12]. Recently, MSCs pre-induced via addition of either β-mercaptoethanol or basic fibroblast growth factor to the cell medium were reported to undergo neuronal differentiation [13,14,15,16]. Therefore, the present study aims to evaluate the effect of a pre-induced MSC-coated cellulose/collagen nanofibrous nerve conduit for facial nerve regeneration in a rat model, both in vitro and in vivo. 

## 2. Results

### 2.1. In Vitro Study

To examine the neuronal pre-induction effect, MSCs were treated with neuronal pre-induction medium for 24 h. Then, the gene expression levels of neuro-progenitor markers, trophic factor, and p21 were measured using RT-PCR. The expression of genes encoding neuro-progenitor markers (CD133, GFAP, Musashi, and Nestin), trophic factors (ANG, BDNF, and VEGF), and p21 was significantly increased (Figure 1). Increases in Musashi, Nestin, GFAP, ANG, and p21 gene expression levels were confirmed using immunocytochemical staining, and consistent results were obtained (Figure 2). Based on these results, the effect of pre-induced MSCs was established. Moreover, scanning electron microscopy showed that the MSCs were well-attached to the lumen of the cellulose/collagen nanofibrous conduit (Figure 3). 

### 2.2. In Vivo Study

#### 2.2.1. Recovery of Vibrissa Fibrillation

No postoperative complications were observed and all rats survived well. As shown in Figure 4, the recovery of vibrissa movement was more prominent in group II (cellulose/collagen nanofiber/neural pre-induced MSC group) than in the control group. The recovery of vibrissa fibrillation was enhanced by time dependent pattern at postoperative weeks 2, 4, 6, and 8 (* *p* < 0.05, Figure 4).

#### 2.2.2. Macroscopic Observation

In all the three groups, the nerve gap was successfully regenerated with a new bridge in all the rats, and degradation of the cellulose/collagen nanofiber conduit was confirmed macroscopically 8 weeks after surgery. No neuroma formation was observed. The gross thickness of the nerve was not significantly different, although both groups I and II showed slightly larger diameter of the nerve than the control group (Figure 5). 

#### 2.2.3. Electrophysiological Studies

As shown in Figure 6, 8 weeks after the application of the nerve conduit, group II (treated with cellulose/collagen nanofiber nerve conduit/neural pre-induced MSCs) showed significantly lower mean threshold of MAP than the other two groups. The mean threshold of MAP of group I was significantly lower than that of the control group (Figure 6). 

#### 2.2.4. Histopathological Studies

As shown in Figure 7A, both groups I and II (treated with the MSC-coated nerve conduit) exhibited larger axons than the control group. The surrounding myelin sheaths, produced by Schwann cells, were also more distinct and thicker in the groups treated with the MSC-coated nerve conduit than in the control group. Additionally, group II (cellulose/collagen nanofiber nerve conduit/neural pre-induced MSCs) exhibited improved morphological parameters compared with group I. H&E and neurofilament immunostaining (Figure 7B) revealed smaller axonal diameters in the regenerated nerve fibers of the control group. Moreover, Luxol fast blue staining revealed thinner myelin sheaths in the nerve fibers, and S100 immunostaining showed fewer and smaller Schwann cells surrounding the regenerated nerve fibers in the control group compared with those in groups I and II. Therefore, treatment with the MSC-coated nerve conduit alleviated this poor regeneration effect observed in the control group, as shown in Figure 7C. The treatment with neural pre-induced MSCs enhanced the morphological parameters of nerve fiber regeneration by 13–34%. These findings suggest the neuro-regenerative ability of MSC treatment against facial nerve injury due to trauma. 

As shown in Figure 8, the axon counts of three groups were similar (n = 255), but gross size of axon is higher in the pre-induced MSC group than other groups. The thickness of myelin sheath is higher in pre-induced MSC group than other groups. The Kruskal-Wallis test showed significant differences between the three groups (** *p* = 0.0036). 

## 3. Discussion

In the present study, rat facial nerve regeneration was significant in group II. The additional coating of pre-induced MSCs in the cellulose/collagen nanofibrous conduit significantly improved the regeneration parameters. It is proposed that nerve regeneration can be enhanced using an abundant source of collagen, which is an important extracellular matrix component in the nanofiber layer of this conduit. 

Here, a cellulose nanofiber was coated with collagen. Among similar molecules in the extracellular matrix, collagen, gelatin, and fibers are commonly used in combination with biomaterials for tissue engineering owing to their enhancing effect on cell attachment, proliferation, and differentiation. In fact, collagen coating of nanofibers stimulates cell proliferation and tissue-specific gene expression [17]. 

The advantages of MSCs include their easy isolation from various tissues and differentiation into diverse cell types, including muscle, nerve, liver, skin, bone, and adipose cells [18]. MSCs have been widely studied and applied in regenerative medicine, such as in the regeneration of muscle and the nervous system. Particularly, MSCs can be used to regenerate both the central and peripheral nervous systems after injury, damage, or dysfunction. In the current study, we first pre-induced MSCs into neuronal-like cells through incubation of MSCs in pre-induction media containing β-mercaptoethanol and bFGF. However, long-term differentiation or long-term exposure of cells to β-mercaptoethanol increases the expression of stress- and shock-related proteins and decreases the number of viable cells [19]. In the present study, we confirmed the pre-differentiated MSCs by previous Joe et al.’s report [20]. They reported the various neuronal progenitor markers such as CD133, GFAP, Musashi, and Nestin, as well as neurotrophic factors, including ANG, BDNF, and VEGF. Another important characteristic of MSCs is their proliferation; however, pre-induced MSCs showed increased expression of p21, which is involved in cell cycle arrest, suggesting that the proliferation of MSCs here was arrested and that the MSCs were committed to neuronal differentiation. Furthermore, for successful transplantation of pre-induced MSCs, the MSCs were coated into a biologically degradable cellulose/collagen nanofibrous nerve conduit and pre-induced using pre-induction media. The pre-induced MSC-coated nerve conduit was then transplanted to injured sites in rats. 

In this study, we fabricated a cellulose/collagen nanofibrous nerve conduit coated with pre-induced MSCs for application in a defective facial nerve. Although the three groups showed similar gross appearance of the regenerated nerve, group II exhibited the highest recovery of vibrissa fibrillation and action potential threshold. The highest degree of recovery based on histological findings was also observed in group II. The transplantation of MSC or neurodifferentiated MSC enhanced the regeneration of Schwann cells and thickness of myelin sheath [21,22]. In the present study, although we did not observe the significantly increased axon counts between three groups, but the thickness of myelin sheath was enhanced regeneration in the MSC treated group I and II compared to control group. The presence of S100 positive cells in the regenerated nerve fibers indicated that Schwann cells may have played a central role in the myelin sheath and axonal growth observed. Schwann cells aid functional recovery in injured peripheral nerves by promoting axonal regeneration and myelin rebuilding [23]. Schwann cells are necessary for effective nerve regeneration; in response to injury, they partially “de-differentiate” re-starting the production of developmental genes that assist nerve repair [24,25]. In congruence with the current study, mesenchymal stem cells also derived from human umbilical cords, have also been successful in enhancing nerve regeneration in transected sciatic nerves of adult rats [26]. Moreover, Schwann cells differentiated from bone marrow-derived mesenchymal stem cells have also successfully repaired spinal cord injury in rats [27]. However, whether the Schwann cells found in groups I and II in the current study are exogenous or endogenous—remain to be investigated. Further studies are also needed to identify the specific underlying mechanisms involved in the regenerative properties exhibited by the pre-induced MSC laden nerve conduit used in the current study.

Although direct approximation by end-to-end anastomosis has a limitation of possible misalignment, the nerve conduit provided good alignment and reduced tension and tension-induced ischemia at the repair site. Moreover, the application of a nerve conduit does not need sutures, thus avoiding the negative inflammatory effect of sutures. However, in case of the nerve gap is over than 5 mm, incomplete regeneration can be occurred. To overcome this incomplete regeneration, different growth factors, cells, and modifications of the internal framework should be examined [28,29,30,31]. 

## 4. Materials and Methods

### 4.1. Fabrication of the Nerve Conduit

Cellulose acetate (density = 1.3 g/cm^3^, Mn = 30,000 g/mol) was obtained from Sigma–Aldrich (St. Louis, MO, USA) for electrospinning. Porcine collagen type I was purchased from MSBIO, Inc. (South Korea). To dissolve cellulose into 20 wt% solution, a solvent mixture of acetone and dimethylformamide (1:1) was used. A cylindrical cellulose/collagen scaffold was fabricated using an electrospinning process followed by a collagen coating process. As shown in Figure 1, the electrospinning instrument consisted of a syringe pump, a high-voltage direct current, and a rotating collector. The cellulose solution flow rate was 0.5 mL/h, and the voltage was 13 kV. The rotating collagen speed was approximately 0.3 m/s, and the distance between the nozzle tip and the collector was 8 cm. The electrospinning deposition time was 2 h. After deposition of the cellulose fiber, a nanofibrous conduit was detached from the rotating collector. The nanofibrous conduit was then dipped into the 0.1 wt% collagen solution for 2 h and dried at room temperature (RT) for 24 h. Next, the collagen was immersed in a 50 mM 1-ethyl-(3-3-dimethylaminopropyl) hydrochloride (EDC) solution in 95% ethanol to crosslink the collagen for 1 h at RT. To remove the EDC solution, the cellulose/collagen nanofibrous conduit was washed three times in a 0.1 M sodium hydrogen phosphate solution and three times in deionized water (Figure 9).

### 4.2. Scanning Electron Microscopy of the Cellulose/Collagen Nanofiber Nerve Conduit

The cellulose/collagen nanofibrous conduit and cellulose/collagen MSC-coated nanofibrous conduit were immersed in fresh 2.5% glutaraldehyde fixative solution overnight and then immersed in OSO_4_ solution. After dehydration using critical point drying, platinum sputter coating was performed. The surface of the nerve conduits was analyzed using scanning electron microscopy (FE-SEM, Hitachi, Tokyo, Japan) at the Korea Basic Science Institute. 

### 4.3. Cell Culture and Neuronal Pre-Induction

Human bone marrow-MSCs (hBM-MSCs) were purchased from Cell Engineering for Origin (Korea). These MSCs were cultured in low-glucose Dulbecco’s modified Eagle’s medium (Gibco BRL, NY, USA) supplemented with 10% fetal bovine serum (Gibco, NY, USA) and 1% penicillin and streptomycin (Gibco) at 37 °C and 5% CO_2_ in a humidified atmosphere. hBM-MSCs at the seventh passage (P-7) were then seeded over a cellulose/collagen nanofiber scaffold in 12-well plates at a density of 1.5 × 10^4^ cells/well. The next day, the cells were treated with pre-induction medium as described previously [14,15].

### 4.4. Real-Time Polymerase Chain Reaction (PCR)

RNA was extracted from hBM-MSCs using RNAiso Plus (TAKARA, Japan). cDNA was prepared using a PrimeScript II cDNA Synthesis kit (TAKARA). PCR analyses were performed using Power SYBR Green Mix (Applied Biosystems, Carlsbad, CA, USA) and human sequence-specific primers for the β-actin, (angiopoietin) *ANG*, *BDNF*, *CD133*, *bFGF*, *GFAP*, Musashi gene, Nestin-1, *p21*, and *VEGF* genes (Table 1), which were synthesized by Integrated DNA Technologies, Inc. (Coralville, IA, USA) and Genotech (Daejeon, South Korea).

### 4.5. Immunocytochemistry

hBM-MSCs were grown on coated coverslips and treated with neuronal pre-induction medium for 24 h. Thereafter, the cells were treated with primary antibodies against ANG (1:200), GFAP (1:200), Musashi (1:200), Nestin (1:200), and p21(1:200) (Santa Cruz Biotechnology, CA, USA) for 2 h at RT. Next, the cells were further incubated with secondary antibodies, namely donkey anti-goat immunoglobulin (Ig)G antibody conjugated with Alexa 555 (1:400) for Nestin and donkey anti-rabbit IgG antibody conjugated with Alexa 488 (1:500) for the other markers. All secondary antibodies were applied together with Hoechst 33,342 (1:1000) for 1 h 30 min at RT. After being washed with phosphate-buffered saline, the cells were mounted using the Prolong Gold anti-fade reagent and visualized under a Nikon Eclipse Ti2 fluorescence microscope (Nikon, Tokyo, Japan). Cell images were acquired using a DS-Ri2 digital camera (Nikon). All secondary antibodies, Hoechst 33342, and the anti-fade mounting reagent were purchased from Molecular Probes (ThermoFisher Scientific Korea, Seoul, Korea)

### 4.6. In Vivo Study

#### 4.6.1. Nerve Conduit Application in a Rat Facial Nerve Transection Model

Fifteen adult male Sprague-Dawley rats (6–8 weeks old, weighing 200–250 g; SamtakoBio Korea, Suwon, South Korea) were used in this study. The rats were randomly divided into three treatment groups (n = 10 in each): cellulose/collagen nanofiber (control group), cellulose/collagen nanofiber/MSCs (group I), and cellulose/collagen nanofiber/pre-induced MSCs (group II). Each rat was housed in a separate cage and provided feed and water. They were allowed to adapt to the environment without stress for a week before surgery. This study was approved by the Animal Experimentation Committee (CIACUC2021-S0021). 

A postauricular incision on the left side was performed because the left side allows easy setup for measuring action potential threshold. After identification of the main trunk of the facial nerve using a surgical microscope (Leica co, Wetzlar, Germany), nerve defect was created by transection through a cut in the middle of the main trunk using a microscissor. A 4 mm length cellulose/collagen nanofiber conduit was interposed in this area, and the transected proximal and distal nerve stumps were anchored to the conduit using fibrin glue. A 2 mm gap was thus formed in the main trunk (Figure 10). Finally, the wound was closed using automatic suture. 

#### 4.6.2. Evaluation of Vibrissae Fibrillation 

Vibrissa fibrillation in both groups I and II were recorded for 40 s each using the iPhone video system 2, 4, 6, and 8 weeks after surgery. The frequency of vibrissa fibrillation was analyzed using BORIS (Behavioral observation research interactive software), an animal behavior evaluation software for video/audio coding and live observations. The authors of BORIS are Oliver Friard and Marco Gamba (Department of life sciences and systems biology, University of Torino, Italy). The percentage of the frequency of vibrissa fibrillation (left side: nerve conduit site, right side: normal site) was calculated. The comparison between the three groups at postsurgery 2nd, 4, 6, and 8th week was performed by repeat measure ANOVA test.

#### 4.6.3. Measurement of Threshold of Electrically Stimulated Muscle Action Potential

The facial nerves were re-exposed under general anesthesia using isoflurane inhalation at postoperative week 8. After electrical stimulation was applied to the distal part of the nerve conduit using a monopolar tungsten probe, the threshold of action potential was measured as described previously [32]. In brief, three needle electrodes were inserted percutaneously into the midpoint of the left orbicularis oculi muscle, left orbicularis oris muscle, and superficial muscle layer near the skin (ground needle) to record electrically evoked muscle action potential (MAP) signals. Electrical signals (rectangular current pulse for 0.05 ms) were delivered to the main trunk of the facial nerve using a monopolar stimulating electrode (Xomed-Treace, Jacksonville, FL, USA), which was connected to a pulse generator (A-320D; World Precision Instruments Inc., Sarasota, FL, USA). The distance and direction of the monopolar stimulating probe relative to the facial nerve can be controlled using a micro-manipulator. All MAPs were measured through maximal nerve stimulation. Data were automatically acquired using the lab chart system (PowerLab; AD Instrument, Castle Hill, Australia), which was displayed on a Samsung computer monitor, and then analyzed using the Scope software (AD Instrument). The peak amplitude of the action potential waveform was determined to assess recovery from facial nerve injury. 

#### 4.6.4. Histological Examination Using Hematoxylin and Eosin (H&E), Luxol Fast Blue, and Immunohistochemical Staining for Neurofilament and S-100 

##### Tissue Processing and Histochemical Analysis

Segments of nerve tissue sections in the nerve conduit were carefully dissected and fixed in 4% paraformaldehyde in phosphate-buffered saline or The fixed tissues were processed as per routine, embedded in paraffin, sectioned into 4-μm-thick sections, deparaffinized, and rehydrated using standard protocols. Overall morphology was visualized using routine H&E staining. Myelin was stained using Luxol fast blue. In brief, the rehydrated tissue sections were incubated in 0.1% Luxol fast blue solution overnight at 56 °C, rinsed with 95% ethyl alcohol and distilled water, and differentiated in 0.05% lithium carbonate solution. Subsequently, the sections were dehydrated in a series of alcohol solutions, cleared using xylene, and mounted in a resinous medium. 

### 4.7. Immunohistochemical Analysis

Axonal microtubules and Schwann cells were visualized using immunohistochemical staining with anti-neurofilament and anti-S-100 antibodies, respectively. Briefly, the rehydrated sections were blocked using normal goat serum (Vector ABC Elite Kit; Vector Laboratories) for 1 h, incubated with rabbit anti-neurofilament and anti-S-100 primary antibodies (1:500; Abcam, Cambridge, UK) overnight at 4 °C, reacted with biotinylated goat anti-rabbit IgG (Vector ABC Elite Kit) for 2 h at RT, reacted with the avidin–biotin peroxidase complex (Vector ABC Elite Kit) for 1 h at RT, and finally developed using diaminobenzidine substrate (DAB kit; Vector Laboratories). The relative staining intensities, average positive cell sizes, and average axonal diameters were analyzed using the ImageJ software. The parameters are expressed as the mean ± standard error (n = 3/group).

#### Ultrastructural Findings Using Transmission Electronmicroscopy

After euthanasia, the distal portion of the facial nerve from 3 animals of each group was rapidly excised using Dorco stainless razor blade and immediately immersed in 2.5% glutaraldehyde fixation, tissue samples were washed in phosphate buffer and post-fixed using 1% osmium tetroxide. After serial dehydration using ethanol, nerve samples were then embedded in a mixture of resins (LR white resin). Semi-thin transverse sections were cut at 1 mm distal to the site of regeneration and stained with toluidine blue. Sections were evaluated by light microscope. Ultra-thin sections were cut immediately after the series of semi-thin sections. They were examined using a JEM-2100F field emission transmission electron microscope (JEM-2100F, JEOL Ltd., Tokyo, Japan). The thickness of myelin were obtained using image J. 

### 4.8. Statistical Analysis

Statistical analysis was performed using GraphPad Prism 8.0. The three groups were compared using one-way ANOVA. The recovery of vibrissae fibrillation was performed by repeat measure ANOVA. The thickness of myelin sheath in 3 groups was analyzed using Kruskal-Wallis test. The Mann-Whitney test was used for comparison between two groups. The significance was considered when *p* value is less than 0.05.

## 5. Conclusions

From our results, the neural pre-induced MSC-loaded cellulose/collagen nerve conduit may be helpful for regeneration of facial nerve injury due to trauma. 

## Figures and Tables

**Figure 1 ijms-23-07638-f001:**
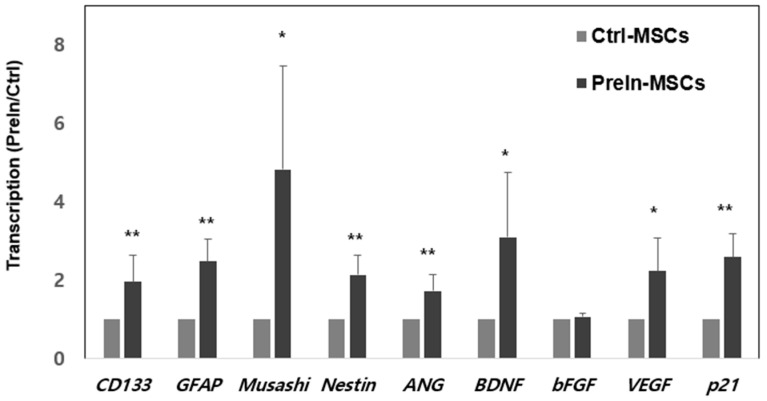
The expression of specific markers was identified by real-time PCR. Transcripts of neuro-progenitor markers (*CD133, GFAP, Musashi,* and *Nestin*), trophic factors (*ANG, BDNF,* and *VEGF*), and *p21* were significantly increased in pre-induced-MSCs (*t*-test, * *p* < 0.05, ** *p* < 0.005, mean ± SD n = 6). Control-MSCs: control-MSCs. PreIn-MSCs: preinduced-MSCs which are incubated with pre-induction media for 24 h.

**Figure 2 ijms-23-07638-f002:**
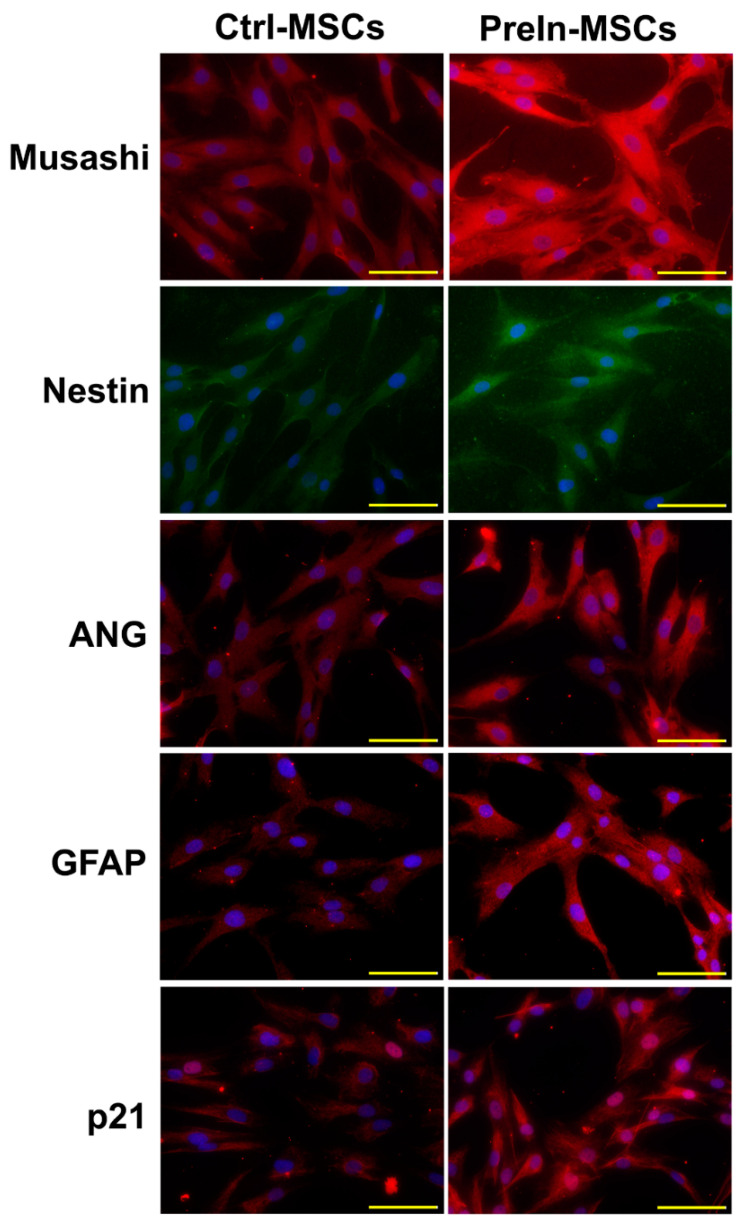
Control (Left panel) and pre-induced MSCs (Right panel) were fixed and stained with specific antibodies, Muisashi, Nestin, ANG, GFAP, and p21, and visualized under fluorescent microscopy (red and green). Blue indicates nucleus. Pre-induced MSCs indicates preinduced-MSCs which are incubated with preinduction media for 24 h.

**Figure 3 ijms-23-07638-f003:**
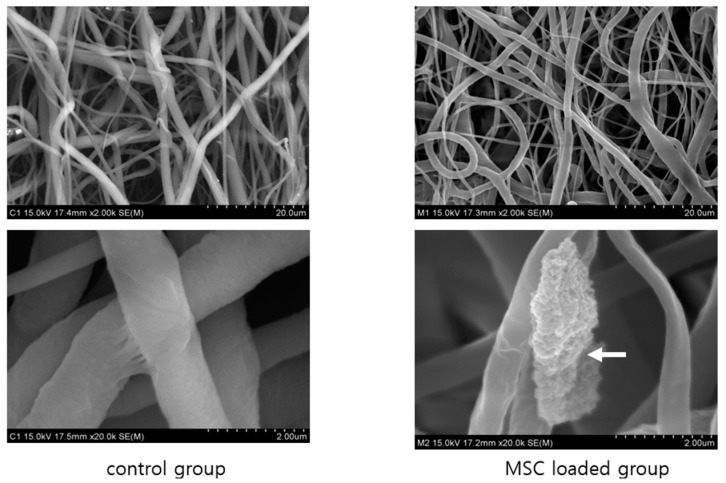
Scanning electron microscopic view of control group and MSC loaded group. Both groups show cellulose/collagen nanofibrous structure (2.0K). The magnified view shows well coated MSCs to the nanofibrous strand (arrow indicates).

**Figure 4 ijms-23-07638-f004:**
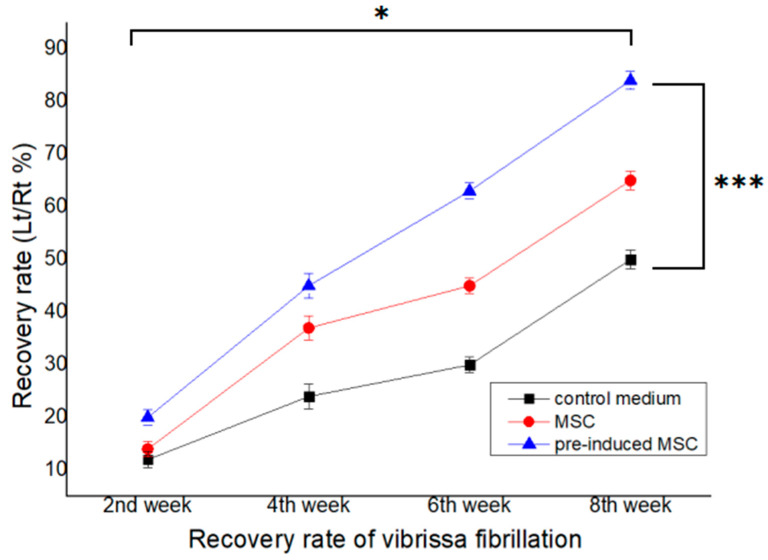
Recovery of vibrissae fibrillation is more pronounced in groups I and II than in the control group. The highest degree of recovery of vibrissae fibrillation is observed in group II (pre-induced MSC). Repeat measure ANOVA, Between time interval; *p* = 0.0226, Between each group; *p* = 0.0047. Values of * *p* < 0.05, *** *p* < 0.001 were considered statistically significant.

**Figure 5 ijms-23-07638-f005:**
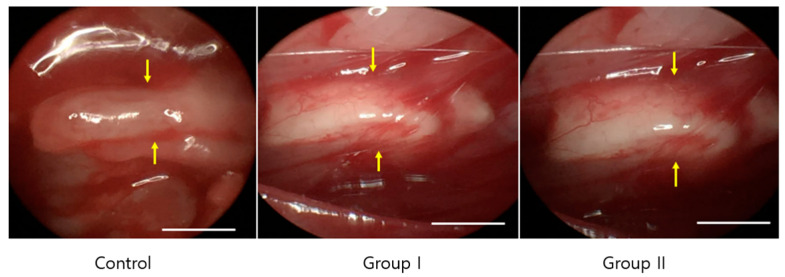
All groups show well-regenerated nerve. Compared with the control group, both the mesenchymal stem cell (MSC) and pre-induced MSC groups show slightly larger diameter of the nerve, but the difference was not significant.

**Figure 6 ijms-23-07638-f006:**
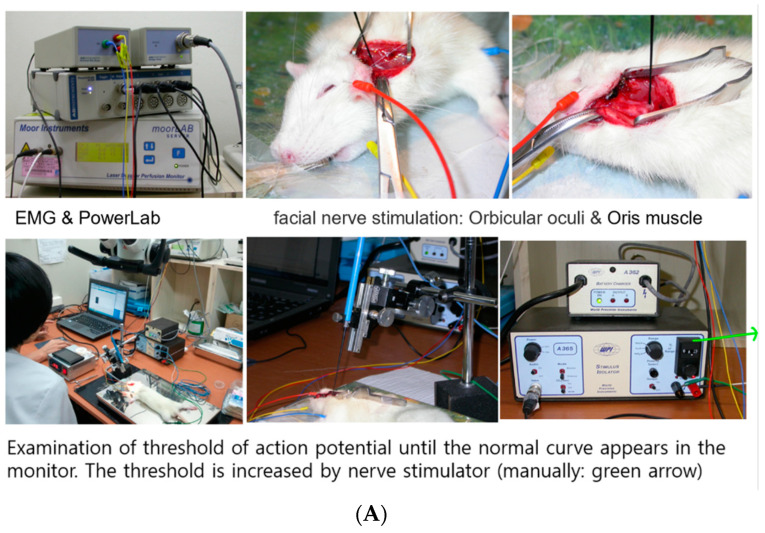
Measurement of threshold of action potential (**A**). The graph of action potential in the lab chart of PowerLab (**B**). The threshold of action potential of each group was compared to normal nerve. Pre-induced MSC coated cellulose/collagen nanofibrous conduit group showed significantly reduced than other groups. One way ANOVA between group, *p* < 0.0001. Values of **** *p* < 0.0001 were considered statistically significant (**C**).

**Figure 7 ijms-23-07638-f007:**
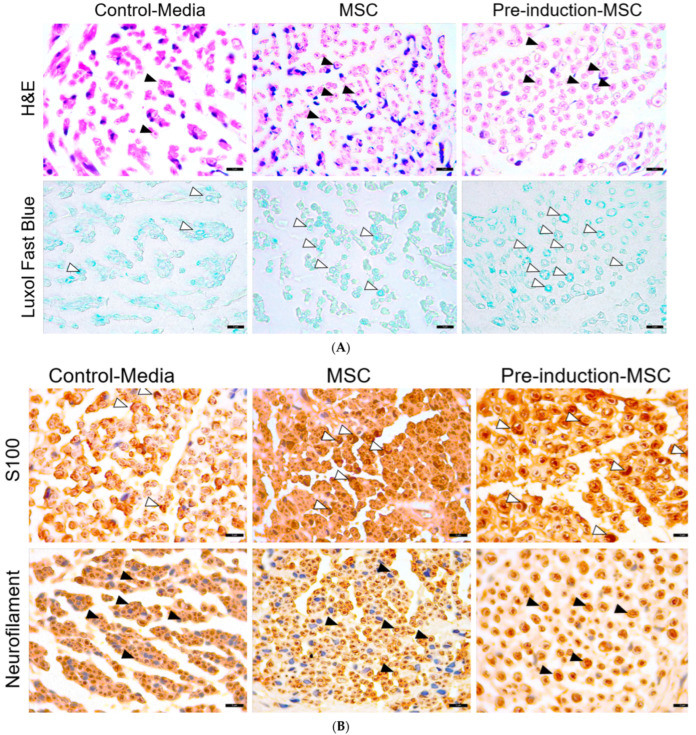
Pre-induced MSC showed better regeneration than other groups. (**A**) both groups I and II (treated with the MSC-coated nerve conduit) exhibited larger axons than the control group. (**B**) H&E and neurofilament immunostaining revealed smaller axonal diameters in the regenerated nerve fibers of the control group. (**C**) treatment with the MSC-coated nerve conduit alleviated this poor regeneration effect observed in the control group. Values of ** *p* < 0.01, *** *p* < 0.001, and **** *p* < 0.0001 were considered statistically significant.

**Figure 8 ijms-23-07638-f008:**
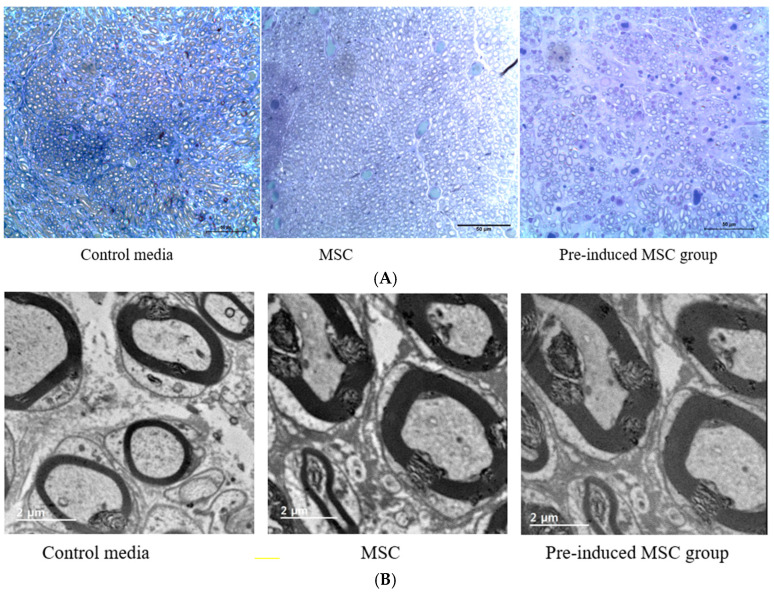
(**A**) The axon counts of three groups were similar (n = 255), but the gross size of an axon is higher in the pre-induced MSC group than other groups. (**B**) The thickness of myelin sheath of the pre-induced MSC group and MSC group show larger than the control media group. (**C**) reveals that thickness of myelin sheath of the pre-induced MSC is highest among three groups. ** *p* < 0.01 were considered statistically significant.

**Figure 9 ijms-23-07638-f009:**
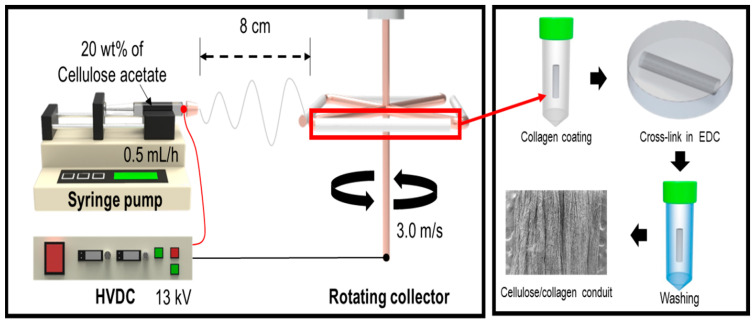
Fabrication schematic of cellulose/collagen nanofibrous conduit scaffold.

**Figure 10 ijms-23-07638-f010:**
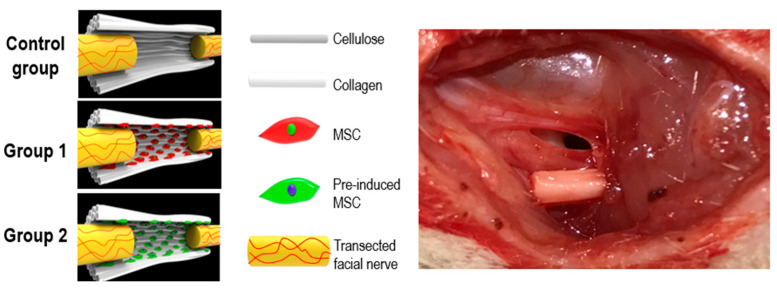
The schematic view of conduit.

**Table 1 ijms-23-07638-t001:** Primers used in the real-time polymerase chain reaction analysis.

Gene	Forward Primer (5′ → 3′)	Reverse Primer (5′ → 3′)
** *CD133* **	GAGCAGGTTGTGTGCTTGGT	GGAAGCACTGGATCTGCTGAA
** *GFAP* **	CAGAAGAGGACACAATGGCG	GTACAGAGCAAGAAGGGCTG
**Musashi**	TCTGTGTAGGGGGACTGTGT	TGAATGGCACAGACCAGGAA
**Nestin**	GATAAGTCAGCCAGGGAGCAG	GACATCTTGAGGTGCGCCAG
** *ANG* **	TGGGCGTTTTGTTGTTGGTC	GGCATCATAGTGCTGGGTCA
** *BDNF* **	ACCCACACGCTTCTGTATGG	GCAGCCTTCATGCAACCAAA
** *bFGF* **	AAAAACGGGGGCTTCTTCCT	ACGGTTAGCACACACTCCTT
** *VEGF* **	AGAAAATCCCTGTGGGCCTT	GTCACATCTGCAAGTACGTTCG
** *p21* **	GTCTTGTACCCTTGTGCCTC	GGCGTTTGGAGTGGTAGAAA
**β-actin**	ATCCGCAAAGACCTGTACGC	TCTTCATTGTGCTGGGTGCC

## Data Availability

The data presented in this study are available on request from the corresponding author.

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
