# Peer review of "Effect of Pre-Induced Mesenchymal Stem Cell-Coated Cellulose/Collagen Nanofibrous Nerve Conduit on Regeneration of Transected Facial Nerve"

_ijms, 2022, doi:10.3390/ijms23147638_

Round 1

Reviewer 1 Report

The authors have shown than Pre-Induced MSC are better to produce an increased regeneration. 

Their data looks clear and their funding is supported by the data. However, it would be more clear and easy to quantify axonal regeneration (and remyelination) if authors could show and quantify electron microscope images in their experimental nerves (or semithin sections and Toluidine blue staining, 1-2um thick sections). 

I would like to highlight that the facial nerve is a peripheral nerve (PNS), and myelin sheaths are made by Schwann cells. Unfortunately, oligodendrocytes are not able to form myelin inside the peripheral nerves (there aren't OPCs inside the PNS). Authors should correct this misunderstood and it will be good to explain a bit of the repair Schwann cell biology. 

In addition, are the Schwann cells penetrating more the Pre-induced MSC conduits? They could check by Immunofluorescence or Electron microscopy. 

Minor comments:

Figure panels should be improved and figure legends as well. 

Figure 2 is missing the scale bar

The study is in rats and the authors show human primers for qPCR (genes in rat are written as Bdnf, Vegf.... and proteins BNDF, VEGF....

Author Response

Reviewer 1

Reviewer’s comment: The authors have shown than Pre-Induced MSC are better to produce an increased regeneration. Their data looks clear and their funding is supported by the data. However, it would be more clear and easy to quantify axonal regeneration (and remyelination) if authors could show and quantify electron microscope images in their experimental nerves (or semithin sections and Toluidine blue staining, 1-2um thick sections). 

Correction: Thank for your comments. Actually we added semithin photos and TME photos. We mearured the thickness of myelin sheath using Image J. But the image quality was not good than we expected. According to your recommendation, we added the photos and description in the Result and Materials and Method section. They were highlighted by yellow color.

Reviewer’s comment: I would like to highlight that the facial nerve is a peripheral nerve (PNS), and myelin sheaths are made by Schwann cells. Unfortunately, oligodendrocytes are not able to form myelin inside the peripheral nerves (there aren't OPCs inside the PNS). Authors should correct this misunderstood and it will be good to explain a bit of the repair Schwann cell biology. In addition, are the Schwann cells penetrating more the Pre-induced MSC conduits? They could check by Immunofluorescence or Electron microscopy. 

Correction:We thank the reviewer for pointing out this mistake. The authors would like to apologize for the lapse in our writing. Indeed, we meant ‘Schwann’ cells instead of ‘Oligodendrocytes’, as the facial nerve is part of the PNS and not the CNS. We have corrected the manuscript to reflect these changes. (see Lines 114 and 118). We have also included some discussion of Schwann cells’ role in nerve injury repair and axonal regeneration. (lines 152 – 164).

Minor comments:

Reviewer’s comment: Figure panels should be improved and figure legends as well. 

Correction: We improved figure panels and legends.

Reviewer’s comment: Figure 2 is missing the scale bar

Correction: We added the scale bar (100µm) and increased pixels on the Figure 2.

Reviewer’s comment: The study is in rats and the authors show human primers for qPCR (genes in rat are written as BdnfVegf.... and proteins BNDF, VEGF....

Correction: In this study we used human bone marrow derived mesenchymal stem cells (hBM-MSCs) for neuronal pre-induction in vitro. Therefore, we evaluated the mRNA level of human specific genes in hBM-MSCs.

I thank the editors and reviewers of the International Journal of Molecular Science for taking their time to review our article. I have made some corrections and clarifications in the manuscript after going over reviewer’s comments.

I hope the revised manuscript will better meet the requirements of the International Journal of Molecular Science for publication. I thank you again for constructive review by the reviewers.

Reviewer 2 Report

This is a very interesting study mainly focusing on the effect of pre-induced MSCs embedded into cellulose/collagen nanofiber conduit as compared to bare or MSC into cellulose/collagen nanofiber conduit.

The authors conducted RT-PCR and immunohistochemistry for the assessing the expression neuro-progenitor markers, trophic factor, and p21 after incubation with neuobasal medium. And the authors also conducted nerve gap model to evaluate the outcome.

However there were some flaws in this study.

In Figure2, the immunohistochemsity staining was not typical and blurred. We recommend the imaging taken by con-focal microscope.

In Figure 3, line 77- misspelling in” gropu” and should be ”group”.

In Figure 4, the recovery of vibrissae fibrillation was assessed in the same animal observed at different time points. The authors misused the statistical method and the data should be measured by repeated ANOVA or general estimation equation.

In Figure 6, please show the typical photography of compound muscle action potential in three groups, respectively.

In Figure 7, it is hard to differentiate the effect from the transplanted cells themselves or the paracrine effects. The authors should require localization of transplanted cells in tissue sections with human-specific in situ hybridization probes.      

Author Response

This is a very interesting study mainly focusing on the effect of pre-induced MSCs embedded into cellulose/collagen nanofiber conduit as compared to bare or MSC into cellulose/collagen nanofiber conduit. The authors conducted RT-PCR and immunohistochemistry for the assessing the expression neuro-progenitor markers, trophic factor, and p21 after incubation with neuobasal medium. And the authors also conducted nerve gap model to evaluate the outcome.

However there were some flaws in this study.

Reviewer’s comment: In Figure2, the immunohistochemsity staining was not typical and blurred. We recommend the imaging taken by con-focal microscope.

Response: In this study we used fluorescent microcopy to capture the immunocytochemical stained hBM-MSCs. As per the reviewer's suggestion, we try to get better images in further studies.

Reviewer’s comment: In Figure 3, line 77- misspelling in” gropu” and should be ”group”.

Correction: We corrected the typo. Line 78. It was highlighted by yellow color.

 Reviewer’s comment: In Figure 4, the recovery of vibrissae fibrillation was assessed in the same animal observed at different time points. The authors misused the statistical method and the data should be measured by repeated ANOVA or general estimation equation.

Correction: We performed the analysis using repeat measure ANOVA, and we attached the new graph with p value. Figure 4.

Reviewer’s comment: In Figure 6, please show the typical photography of compound muscle action potential in three groups, respectively.

Correction: We measured the threshold of action potential using EMG, data acquisition sytem (PowerLab), EMG connected three needle electrodes and tungsten probe for stimulation of facial nerve just like intraoperative facial nerve monitoring system. We measured the normal main trunk, and we increased the amplitude of threshold until normal pattern of graph appears in the monitor using manually control of nerve stimulator. We added the photos in the figure 6.

Reviewer’s comment: In Figure 7, it is hard to differentiate the effect from the transplanted cells themselves or the paracrine effects. The authors should require localization of transplanted cells in tissue sections with human-specific in situ hybridization probes.      

Response: The authors appreciate the comment of the reviewer. Indeed, with the current staining it would be hard to differentiate whether the S100 positive-Schwann cells were from differentiated exogenous human umbilical MSCs or endogenous rat MSCs. Despite this, the authors deem that thecurrent figures are enough to support the main findings of the study which suggests that pre-induced MSCs are more capable of facilitating nerve regeneration compared with non-induced MSCs and non-MSC infiltrated nanofibrous material. The more efficient nerve regenerating-effect brought about by pre-induced MSCs group was adequately supported by the presence of larger S100-positive Schwann cells, coupled with thicker myelin sheaths (as stained by Luxol fast blue) and thicker axons (as stained by Neurofilament). Admittedly, the use of human-specific in situ hybridization probes would be extremely beneficial in future studies that would further explore the underlying mechanisms of the nerve regenerating capabilities of the pre-induced MSC-infused nerve conduits. As such we have added these narrative and suggestions to the discussion section of the manuscript (Lines 153-164).

Round 2

Reviewer 2 Report

The authors had made the changes required for the publication. The manuscript should be published. 

Author Response

Reviewer's comment: The authors had made the changes required for the publication. The manuscript should be published. 

Response: Thank you for your comment. I thank again for your constructive review.